# Transcriptome Analysis Reveals Fruit Quality Formation in *Actinidia eriantha* Benth

**DOI:** 10.3390/plants12244079

**Published:** 2023-12-06

**Authors:** Peiyu Wang, Xin Feng, Jinlan Jiang, Peipei Yan, Zunwen Li, Weihong Luo, Yiting Chen, Wei Ye

**Affiliations:** 1Sanming Academy of Agricultural Sciences, Shaxian 365051, China; 17268144914@163.com (P.W.); 13616957010@163.com (J.J.); 18296793560@163.com (Z.L.); 2The Key Laboratory of Crop Genetic Improvement and Innovative Utilization in Fujian Province (Mountain Area), Shaxian 365051, China; 3Fruit Tree Research Institute of Fujian Academy of Agricultural Sciences, Fuzhou 350002, China; fengxin1506@163.com; 4Institute of Horticultural Plant Bioengineering, Fujian Agriculture and Forestry University, Fuzhou 350002, China; L1224142990@163.com

**Keywords:** *Actinidia eriantha* Benth, fruit quality, organic acid, vitamin C, sugars, transcriptome sequencing

## Abstract

*Actinidia chinensis* Planch. is a fruit tree originating from China that is abundant in the wild. Actinidia eriantha Benth. is a type of *A. chinensis* that has emerged in recent years. The shape of *A. eriantha* is an elongated oval, and the skin is covered with dense, non-shedding milk-white hairs. The mature fruit has flesh that is bright green in colour, and the fruit has a strong flavour and a grass-like smell. It is appreciated for its rich nutrient content and unique flavour. Vitamin C, sugar, and organic acids are key factors in the quality and flavour composition of *A. eriantha* but have not yet been systematically analysed. Therefore, we sequenced the transcriptome of *A. eriantha* at three developmental stages and labelled them S1, S2, and S3, and comparisons of S1 vs. S2, S1 vs. S3, and S2 vs. S3 revealed 1218, 4019, and 3759 upregulated differentially expressed genes and 1823, 3415, and 2226 downregulated differentially expressed genes, respectively. Furthermore, the upregulated differentially expressed genes included 213 core genes, and Gene Ontology enrichment analysis showed that they were enriched in hormones, sugars, organic acids, and many organic metabolic pathways. The downregulated differentially expressed genes included 207 core genes, which were enriched in the light signalling pathway. We further constructed the metabolic pathways of sugars, organic acids, and vitamin C in *A. eriantha* and identified the genes involved in vitamin C, sugar, and organic acid synthesis in *A. eriantha* fruits at different stages. During fruit development, the vitamin C content decreased, the carbohydrate compound content increased, and the organic acid content decreased. The gene expression patterns were closely related to the accumulation patterns of vitamin C, sugars, and organic acids in *A. eriantha*. The above results lay the foundation for the accumulation of vitamin C, sugars, and organic acids in *A. eriantha* and for understanding flavour formation in *A. eriantha*.

## 1. Introduction

As a fruit tree native to China, *Actinidia eriantha* has become one of the most popular in recent years. *A. eriantha* is rich in nutrients, and the fruit contains a large amount of vitamin C (Vc), dietary fibre, and a variety of mineral nutrients [1]. In addition, *A. eriantha* contains approximately 500 to 1379 mg/100 g fresh weight (FW) of Vc; thus, it is not inferior to other varieties of kiwifruit [2]. However, more studies have been performed on the synthesis and metabolism of Vc in *Actinidia chinensis* and *Actinidia deliciosa* than in *A. eriantha*. In addition, *A. eriantha* is popular for its moderate sweetness and acidity and its unique flavour [3]. Sugar and acid components are important fruit quality components of research interest and have important effects on fruit’s flavour. However, research on kiwifruit has focused on *A. chinensis* and *A. deliciosa*, which contain fructose, glucose, and sucrose [4], and quinic acid, malic acid, and citric acid as the primary acid components [5]. To date, the formation of unique *A. eriantha* flavours and the composition of sugars and organic acids have been less studied.

Sweetness is one of the primary factors used to measure fruit quality [6]. The content and composition ratio of sugar compounds such as sucrose, fructose, and glucose determine the sweetness and flavour of fruits [7,8]. For example, fructose is the main sugar substance in apples and pears [9,10], while sucrose is the main sugar substance in watermelon and cantaloupe [11,12]. Furthermore, previous studies have shown that different varieties of kiwifruit have different main sugar substances. In the fruit of kiwifruit ‘Hort 16A’, glucose is the most abundant, followed by fructose, while sucrose is the least abundant [13]. On the other hand, in the fruit of ‘Mitsuko’ kiwifruit, sucrose is the main sugar substance, and the contents of glucose and fructose are relatively low [14]. Sucrose, one of the key substances in plant metabolism, provides an important material foundation for plant growth and development. The synthesis of sucrose in plants is mainly regulated by sucrose synthase (*SUS*), which controls the synthesis of UDP-glucose for sucrose formation [15]. Subsequently, sucrose is converted into fructose and glucose by neutral invertase. Additionally, fructose and glucose can be phosphorylated by fructokinase and hexokinase, forming fructose-6-phosphate and glucose-6-phosphate [16].

Acidity is another important indicator that determines fruit quality [17]. The main organic acids in fruits are usually malic acid, citric acid, tartaric acid, and succinic acid. Additionally, different organic acids have varying levels of acidity, with malic acid being the strongest and tartaric acid being the weakest. The proportion of organic acids to sugar substances in fruit determines the balance of sweetness and acidity [18]. Furthermore, quinic acid and citric acid are the main organic acids found in kiwifruit ‘Ganmi 6’, yellow-fleshed kiwifruit ‘Jinshi 1’, and *A. chinensis* [19,20,21]. However, some studies also suggest that malic acid is a major component of organic acids in kiwifruit [22,23]. Moreover, malic acid and citric acid, as key intermediates in the tricarboxylic acid cycle, also participate in plant respiration and energy metabolism [24]. The types and levels of organic acids in fruits play a crucial role in determining the quality and unique flavour of horticultural crops.

Vc is an essential nutrient for the human body that cannot be synthesized endogenously and thus needs to be obtained from external sources. There are four pathways for the synthesis of Vc in plants, namely, the L-galactose, D-galactose, L-gulose, and myo-inositol pathways [25,26,27,28]. Additionally, plants have a Vc recycling pathway. Previous studies have shown that the majority of plants use the L-galactose and D-galactose pathways as the main routes of Vc synthesis. Kiwifruit also predominantly synthesizes Vc via these two pathways. However, there was significant variation in the Vc content among the different kiwifruit varieties. The fruits of *A. deliciosa* ‘Hayward’ contain 65.5 mg/100 g FW of Vc. The Vc content of the fruit of *A. deliciosa* ranged from 29 mg/100 g FW to 80 mg/100 g FW. The Vc content of the fruit of sanuki gold was more than three times that of *A. deliciosa* ‘Hayward’. In *Actinidia arguta* fruit, the Vc content ranges from 37 to 185 mg/100 g FW [29,30].

As a technology to study the transcription of all genes in cells at the overall level, transcriptome sequencing has extensive applications in plant function research. Transcriptome sequencing was used to identify many genes related to flowering and fruiting in rabbit-eye blueberry (*Vaccinium ashei*) [31]. Using transcriptome sequencing, *Lycium ruthenicum* provides the basis for functional and molecular biology studies on its fruit colouration [32]. Transcriptome sequencing of different stages of fruit development helped to elucidate the molecular mechanism of *Pyrus bretschneideri* Rehd fruit ripening [33]. In addition, transcriptome techniques have been used in fruit trees such as *Prunus armeniaca* L. [34,35] and *Rosa roxburghii* [36]. *A. chinensis* Planch. is a perennial deciduous fruiting woody vine of *Actinidia, Actinidiaceae. A. eriantha* is a unique germplasm resource of kiwifruit in China, and more than 300 accessions have been recorded and evaluated. At present, most *A. eriantha* are found in the wild. *A. eriantha* ‘White’, *A. eriantha* ‘Shanong 18’, and *A. eriantha* ‘Ganmi 6’ are the dominant *A. eriantha*, and only one hybridized cultivar, *A. eriantha* ‘Jinyan’, has been registered. *A. eriantha* fruits are rich in Vc, minerals, and a variety of amino acids, with unique flavours [37]. Therefore, the synthesis and accumulation of vitamins, sugars, and organic acids during the growth and development process play a crucial role in the fruit quality and its unique flavour. Previous transcriptomic studies on kiwifruit have analysed its microsatellite markers for population genetics studies and the new formation mechanisms of the early maturity traits in kiwifruit [38]. However, systematic analysis and determination of the Vc, sugar, and organic acid contents in kiwifruit have not been reported. Therefore, this study used transcriptome sequencing technology to analyse samples from different stages of fruit growth and development of *A. eriantha* [39]. The transcriptome results showed that many differentially expressed genes (DEGs) were identifiable during the development of *A. eriantha* fruits. Similarly, we identified 213 and 207 core upregulated and downregulated genes among the DEGs. The Gene Ontology (GO) enrichment showed that the 213 core upregulated DEGs were mainly enriched in hormones, sugars, organic acids, and many organic matter metabolism pathways, and the 207 core downregulated DEGs showed a greatly reduced response to light during *A. eriantha* ripening. To further understand the fruit flavour formation of *A. eriantha*, we further constructed a synthesis pathway of sugar, organic acids, and Vc during fruit development in *A. eriantha*. Furthermore, we measured the contents of sugars, organic acids, and Vc during fruit development. Finally, we verified the relationship between genes and Vc, sugar, and organic acid using real-time fluorescence quantitative PCR (RT–qPCR). This study provides a reference for understanding the formation of thw unique flavours in *A. eriantha*, which can be used for genetic breeding and variety improvement.

## 2. Results

### 2.1. Transcriptomic Analysis of A. eriantha 

Transcriptome sequencing was performed using nine samples of *A. eriantha* at different stages of growth and development (Figure 1A). The sequencing yielded 57.60 Gb of raw data, with a Qphred 30 (Q30, the probability of correct sequencing for one base is 99.99%) of all data greater than 93.35% (Appendix A). Further low-quality read data were removed using the Fsatp software with the default parameters and aligned with the *A. eriantha* genome, and the alignment rate was 82.37% to 84.11% (Appendix A). The results indicate a high quality of transcriptome data for sequencing. In addition, the results of principal component analysis (PCA) showed that the three biological replicates of the three samples clustered together and that principal component 1 (PC1) explained 89.8% of the observed variation, representing the difference between the samples (Figure 1B).

To gain insight into the dynamic changes in gene expression during *A. eriantha* ripening, we analysed the fragments per kilobase of exon model per million mapped fragments (FPKM) of data from three stages. We identified 1218, 4019, and 3759 upregulated differentially expressed genes and 1823, 3415, and 2226 distinct downregulated differentially expressed genes in S1 vs. S2, S1 vs. S3, and S2 vs. S3, respectively (Figure 2A–D). Moreover, further analysis of these DEGs revealed that the upregulated DEGs included 213 core genes (Figure 2E), and the downregulated DEGs included 207 core genes (Figure 2F). Therefore, we further performed a GO enrichment analysis on these core genes. The 213 core upregulated DEGs were mainly enriched in hormones, sugars, organic acids, and many organic matter metabolism pathways (Appendix A), indicating an important role in the accumulation and metabolism of hormones and organic matter during the maturation of *A. eriantha*. However, the GO enrichment of 207 core downregulated DEGs showed a greatly reduced response to light during *A. eriantha* ripening (Appendix A).

### 2.2. Analysis of the Vc Synthesis Pathway in A. eriantha

Vc is one of the important factors in *A. eriantha* quality. To understand the dynamic changes in Vc content during fruit ripening, we constructed the Vc anabolic pathway (Figure 3A) of *A. eriantha*. The galactose and inositol pathways make limited contributions to Vc synthesis in plants. Therefore, in this paper, the expression profile of Vc synthesis-related genes in *A. eriantha* was constructed based on the L-galactose pathway, galacturonate pathway, and recycling pathway. The results showed that the expression patterns of genes in the L-galactose pathway, galacturonate pathway, and recycling pathway were consistent during the development of *A. eriantha*. They are highly expressed in early and middle fruit development, while gene expression decreases after fruit maturity. Furthermore, we determined the Vc content (Figure 3B) at the three stages of *A. eriantha* development. The results showed that the Vc content was significantly reduced after fruit maturation compared with early fruit development. However, the expression trends of five, two, and six genes in the L-galactose pathway, the galacturonic acid pathway, and the recycling pathway, respectively, showed inconsistencies with the pattern of Vc accumulation in Chinese kiwifruit (Figure 3A,B). Therefore, to further elucidate the correlation between Vc accumulation and gene expression in *A. eriantha*, we conducted RT–qPCR analysis of these genes in the above three pathways. In the L-galactose pathway, we screened five genes from the glucose-6-phosphate isomerase (*PGI*), mannose-6-phosphate isomerase (*PMI*), GDP-D-mannose pyrophosphorylase (*GMP*), and GDP-L-galactose phosphorylase (*GGP*) gene families (Figure 4A). Among the five genes, the *PGI* (*DTZ79_10g09380*) and *GMP* (*DTZ79_13g11340*) expression levels increased during fruit development. However, the expression of *PGI* (*DTZ79_10g12530*) and *PMI* (*DTZ79_02g10970*) decreased during fruit development, while *GGP* (*DTZ79_29g10040*) expression was decreased at S2 and peaked at S3. In addition, in the galacturonate pathway, the *D-galacturonate reductase* (*GalUR*) (*DTZ79_13g02680*) and *aldonolactonase* (*Alase*) (*DTZ79_03g07860*) expression levels decreased with fruit development (Figure 4B). In the Vc recycling pathway, we selected six genes (Figure 4C). The expression of *L-ascorbate peroxidase* (*APX*) (*DTZ79_25g10570*) peaked at S2, and the expression of *APX* (*DTZ79_05g07210*) increased with fruit ripening. Two genes of the L-ascorbate oxidase (*AO*) (*DTZ79_27g07430* and *DTZ79_27g07610*) and two genes of the *monodehydroascorbate reductase* (*MDHAR*) (*DTZ79_27g01630* and *DTZ79_27g11730*) decreased in expression as the fruit matured. The above results indicate that Vc in *A. eriantha* accumulates significantly at the early stages of fruit development while the accumulation pattern of Vc was basically consistent with the expression pattern of the gene.

### 2.3. Analysis of the Sugar Metabolic Pathway in A. eriantha

Sugar compounds are one of the main factors of kiwifruit’s flavour and fruit quality. To explore the anabolism of sugars in *A. eriantha*, several major sugar synthesis pathways were constructed in this study. The results showed that many genes involved in the synthesis of sucrose, fructose, and glucose were highly expressed during fruit ripening (Figure 5A). Among the 11 members of invertase (*A/N-INV*) that regulate the conversion of sucrose into fructose and glucose, 8 were highly expressed during fruit ripening. The above results indicate that the accumulation of sugar during the ripening process of *A. eriantha* shows a positive correlation with fruit ripening. In addition, we further analysed the contents of total sugar, sucrose, fructose, and glucose in the flesh of *A. eriantha* during the three stages of fruit ripening (Figure 5B). During the ripening process of *A. eriantha*, the contents of total sugar, sucrose, fructose, and glucose significantly increased and reached their peak at fruit maturity. To further validate the relationship between genes and sugar accumulation, we conducted RT–qPCR analysis of the expression patterns of six *A/N-INV* genes, two *hexokinase-like* (*HKL*) genes, and one *fucokinase-like* (*FRK*) gene during fruit development. Among the six *A/N-INV* genes, three members showed an increase in expression levels as the fruit matured, while three members showed a decrease in expression levels as the fruit matured. This result suggests that different members are involved in the synthesis of *A. eriantha* sugars at different stages. Moreover, the expression levels of *HKL* and *FRK* also decreased with fruit maturation (Figure 6). This suggests a decrease in the phosphorylation of sugars during the later stages of *A. eriantha* fruit development. The above results suggest that increased expression of some sugar synthesis genes and decreased expression of sugar phosphorylation genes contribute to sugar accumulation in *A. eriantha*.

### 2.4. Analysis of the Organic Acid Metabolism Pathway in A. eriantha

Organic acids, as flavouring substances in fruits, play a crucial role in the formation of special flavours in fruits. To investigate the accumulation of organic acids in *A. eriantha*, we constructed a metabolic pathway of organic acids that showed a downward trend (Figure 7A). NAD-dependent malic enzyme (*NAD-ME*), as an important gene for the synthesis of malic acid, also has a high expression level during the ripening of *A. eriantha*. However, the accumulation of malic acid mainly depends on the transport of AL-activated malate transporter (*ALMT*) to the vacuoles for storage. However, the expression level of *ALMT* is relatively low during the maturation of *A. eriantha*. Therefore, the high expression of *NAD-ME* does not promote the accumulation of malic acid, and malic acid may be involved in other metabolic pathways. In addition, we further measured the contents of titratable acid, malic acid, and citric acid in the flesh of *A. eriantha* during the three stages of fruit ripening. The results showed that during the ripening process of *A. eriantha*, the content of organic acids significantly decreased (Figure 7B). The expression of *NAD-ME* differed from the accumulation pattern of organic acids. Therefore, according to transcriptome data, we selected two *NAD-ME* members with the largest expression differences during fruit ripening and carried out RT–qPCR analysis (Figure 7C). The results showed that the expression levels of these two *NAD-ME* genes significantly decreased during fruit ripening. The above results indicate that the content of organic acids decreases during the maturation process of *A. eriantha* and that the decrease in organic acid content is related to gene expression.

## 3. Discussion

Kiwifruit is a perennial fruiting woody vine originating from China. Although there are abundant wild resources of kiwifruit in China, there are significant differences in quality and shape between different varieties of kiwifruit. Therefore, it is very important to obtain a deeper understanding of the quality of kiwifruit and the formation of its unique flavour. *A. eriantha*, with a high Vc content, unique fruit flavour, and other *characteristics,* has become the most popular variety on the market together with *A. chinensis*, *A. deliciosa*, and *Actinidia arguta* [40].

Sugar and acid, as some of the most important qualities of fruits, have always been a research hotspot. During the development process of *A. eriantha* fruits, total sugar, sucrose, fructose, and glucose continue to accumulate. The accumulation pattern of sugar compounds in kiwifruit is similar to that in fruits such as dates, pears, and watermelons [41,42,43]. Furthermore, sucrose is the main carbohydrate compound in *A. eriantha*, with a proportion of approximately 80%, while the contents of fructose and glucose are similar to those in *A. arguta* [44]. The contents of sucrose, fructose, and glucose in *A. chinensis* ‘Hort 16A’ are similar [14]. The contents of titratable acid, malic acid, and citric acid in *A. eriantha* fruit decreased with ripening, and the content was the lowest at ripening. The content of citric acid is much higher than that of malic acid, which is also true for *A. chinensis* ‘Hort 16A’ and *A. arguta* [14,44]. In addition, the accumulation pattern of organic acids in kiwifruit is opposite to that in apples, peaches, plums, and loquats [41,45,46]. Therefore, even in different varieties of kiwifruit within the same species, there are significant differences in the accumulation patterns of sugar compounds and organic acids. The synthesis and metabolism of carbohydrate compounds and organic acids are key to fruit quality, and they are usually not regulated by a single gene. For example, previous studies on sugar and organic acids and related genes in fruits such as watermelon [11], peaches [42], and citrus [47] have shown that a single gene has a limited contribution to the overall accumulation of sugar and organic acids in the fruit. Therefore, we constructed a pathway for the synthesis of sugars and organic acids in *A. eriantha* and studied the expression profiles of genes involved in sugar and organic acid synthesis in *A. eriantha* during fruit development. Our results show that the genes involved in sugar and organic acid synthesis during *A. eriantha* fruit development have a basically similar expression trend with related sugars and organic acids. Furthermore, using transcriptome sequencing and RT–qPCR we found that the accumulation of organic acids in Trichotomis is directly related to organic acid synthesis genes. The three key gene families are the *CS* gene family, *NAD-ME* gene family and the *ALMT* gene family. *ALMT* and *NAD-ME* regulate the accumulation of malic acid in peaches, while *SUS* and *A/N-INV* participate in the accumulation of carbohydrates, which is similar to that in *A. eriantha* [48,49]. In the Chinese dwarf cherry (*CeraSUS humilis*, Bge.), key genes for the synthesis of organic acids and sugars are similar to those of *A. eriantha* [50]. In addition, similar studies in oriental melon [51], sweet cherry [52], and other fruit trees showed similar results to *A. eriantha*. Together, these results suggest that the accumulation patterns of sugars and organic acids may differ from those of other species, but the key genes are basically the same.

Vc is a necessary substance for the metabolism and growth of all Viridiplantae, and kiwifruit is one of the fruits with the highest Vc content on the market. This study shows that the content of Vc in *A. eriantha* fruit gradually decreases with fruit development. The accumulation pattern of Vc in *A. eriantha* is similar to that reported in previous studies on other kiwifruit species [53,54,55]. *A. eriantha* accumulates a large amount of Vc at the early stages of fruit development in the same way as apples [56] and peaches [57]. In addition, Vc is regulated by four linear and cyclic pathways. These pathways include dozens of enzymes that directly or indirectly participate in the metabolic reaction of Vc [26,27,28,58]. Furthermore, cloning and genetic transformation experiments were conducted on key genes in the Vc synthesis pathway, and no significant increase in Vc content was found [59,60,61,62,63], indicating that Vc is a multigene-, multinetwork-regulated substance. Subsequently, we constructed a pathway for the synthesis of Vc in *A. eriantha* and analysed the expression patterns of the key genes involved. The results showed that most of the genes involved in Vc synthesis were highly expressed at the early and middle stages of *A. eriantha* fruit development, and their expression levels decreased during fruit ripening. The specific expression of these genes is significantly correlated with fruit development, indicating that the development of *A. eriantha* fruit may be an important signal regulating gene expression. As a rate-limiting enzyme in the L-galactose pathway, *GGP* has a critical role in Vc synthesis in higher plants [64,65]. Numerous studies have shown that the expression level of *GGP* is closely related to the Vc content in plants and in kiwifruit [59,64,66,67,68]. The galacturonate pathway, as the second discovered Vc synthesis pathway in plant species, has also been shown to play a role in plant Vc synthesis [58,69]. Although it is not the main Vc synthesis pathway in plants, it is also a pathway for Vc in plants and is closely related to the synthesis of Vc [70,71]. In this study, the L-galactose pathway and the galacturonate pathway were constructed, and the key enzymes were found to be highly correlated with the Vc accumulation pattern in *A. eriantha*, reflecting the important role of the L-galactose pathway and the galacturonate pathway in the Vc synthesis of *A. eriantha* fruit. The pathway of vitamin synthesis and metabolism in higher plants is very complex. Understanding the accumulation and metabolism of Vc in *A. eriantha* can help us fully utilize the germplasm resources of kiwifruit. This study utilized high-throughput sequencing and molecular biology experiments to preliminarily explore the genes of *A. eriantha* and their function in Vc synthesis.

In conclusion, the transcriptome sequencing technique was used to identify gene expression during *A. eriantha* fruit development, providing important information for studying the dynamic changes in gene expression and helping identify key genes that regulate *A. eriantha* fruit development. In addition, the data can be used to identify specific functions of new genes and functional genomes, providing valuable information for the molecular breeding and germplasm resource development of kiwifruit and laying the foundation for genetic research on *A. eriantha*.

## 4. Materials and Methods

### 4.1. Plant Material

In this study, the wild *A. eriantha* materials were collected at the kiwifruit base of Xiamao Town, Shaxian District, the Sanming Research Institute of Agricultural Sciences. According to many years of cultivation observations, the whole *A. eriantha* fruit development process takes approximately 210 to 240 days. For this reason, the samples were collected at 180 days (S1), 210 days (S2), and 240 days (S3) after flowering; 10 to 20 fruits were picked each time, the flesh of the peeled fruits was taken as the experimental material, and the experiment was repeated 3 times. Samples were snap-frozen in liquid nitrogen and stored in a −80 °C refrigerator.

### 4.2. cDNA Library Construction and Transcriptome Sequencing

For more than 10 *A. eriantha* from the same period, the samples were ground, and the experiment was repeated with 3 biological replicates. The total RNA was subsequently extracted from the *A. eriantha* samples at different developmental stages using TRIzol^TM^ RNA extraction reagent (Thermo Fisher, 15596026, Waltham, MA, USA) according to the instructions. The RNA concentration of the samples was measured using a NanoDrop 2000 (Thermo Fisher Scientific, Wilmington, DE, USA), and the RNAn integrity was further assessed using an RNA Nano 6000 from the Agilent 2100 bioanalyzer system (Agilent Technologies, Santa Clara, CA, USA). One microgram of RNA from each sample was used in the NEBNext Ultra^TM^ RNA Library Prep Kit (New England BioLabs, Ltd., Ipswich, MA, USA) to construct cDNA libraries of samples from different stages of kiwifruit development. Finally, transcriptome sequencing was performed using double-end sequencing on the Illumina platform.

### 4.3. Transcriptome Data Analysis

Raw data from transcriptome sequencing were subjected to data quality control using the Fastp.0.23.4 software [72]. The filtered data were aligned with the kiwifruit genome sequence (https://kiwifruitgenome.org/, accessed on 23 August 2023). Subsequently, the results of HISAT2 [73] were compressed and sorted using the SAMtools.1.18 software [74]. Then, the FPKM value for each gene was calculated using the FeatureCounts script 2.16.0 [75]. Finally, we constructed the expression level of the genes in each sample into an expression matrix using the Trinity.2.15.1 software [76].

### 4.4. Differential Gene Analysis and Enrichment Analysis

Differential genes between the different samples were obtained using the DESeq2 R package calculation. DESeq2 provides statistical routines for determining differential expression in digital gene expression data using a model based on the negative binomial distribution. The resulting *p* values were adjusted using the Benjamini–Hochberg approach for controlling the false discovery rate. Genes with an adjusted *p* value < 0.01 and |log2 fold change| > 1 found using DESeq2 were assigned as differentially expressed. In addition, using the GO and KEGG databases, functional enrichment and channel significance enrichment were performed for differential genes between the selected samples using the ClusterProfiler R package 4.10.0 [77].

### 4.5. Vc, Sugar and Organic Acid Determination

An ascorbic acid (ASA) content assay kit (Solarbio^®^ BC1230, Beijing, China) was used to determine the Vc content. For each sample, a total of 0.1 g of the sample was ground well, 1 mL of extract was added, the sample was centrifuged, the supernatant was removed, and the microplate reader wavelength was adjusted to 534 nm for index determination. The glucose content was tested using a Glucose Content Assay Kit (Solarbio^®^ BC2505, Beijing, China), a 0.1 g tissue sample was weighed, 1 mL of distilled water was added, and the sample was ground in a 95 °C water bath for 10 min. After cooling, the sample was subjected to centrifugation, the supernatant was removed, and the microplate reader wavelength was adjusted to 505 nm for index measurement. For sucrose content detection, the Plant Sucrose Content Assay Kit (Solarbio^®^ BC2465, Beijing, China) was utilized. First, 0.1 g of the sample was weighed and ground at normal temperature, followed by the addition of 0.5 mL of extract with proper grinding. The sample was then quickly transferred into a centrifuge tube, placed in an 80 °C water bath for 10 min, and, after cooling, the supernatant was removed. Then, 2 mg of the reagent was added and decolouring was conducted at 80 °C for 30 min, followed by the addition of 0.5 mL of extract. The supernatant was removed, and the enzyme wavelength was adjusted to 480 nm for index determination. For testing the fructose content, the Plant Tissue Fructose Content Assay Kit (Solarbio^®^ BC2455, Beijing, China) was used. Approximately 0.1 g of the sample was weighed, and 1 mL of extract was added, followed by proper grinding. The sample was then placed in an 80 °C water bath for 10 min, and then was centrifuged. The supernatant was removed, followed by the addition of 2 mg of reagent and decolouration at 80 °C for 30 min. Then, 1 mL of extract was added followed by centrifugation at the standard temperature, with the microplate wavelength adjusted to 480 nm for index determination. The total sugar content test was performed using the Total Carbohydrate Content Assay Kit (Solarbio^®^ BC2715, Beijing, China). For citric acid content detection, the Citric Acid (CA) Content Assay Kit (Solarbio^®^ BC2155, Beijing, China) was used, with 0.1 g of tissue weighed, to which 1 mL of reagent was added for ice bath homogenization, followed by centrifugation. The supernatant was placed on ice, and the microplate reader was adjusted to 545 nm for index determination. For malic acid content detection, the Malic Acid Content Assay kit (Solarbio^®^ BC5495, Beijing, China) was used, weighing out a 0.1 g tissue sample, adding 1 mL of extract, and conducting ice bath grinding of the sample. The sample was then centrifuged to produce the supernatant, and the microplate reader was adjusted to 450 nm for index determination. The titratable acid content was tested according to the international standard GB12293-90《Fruit and vegetable products -Determination of titratable acidity》.

### 4.6. Quantitative Analysis

The total RNA was extracted at the 3 stages of kiwifruit development using the TRIzol^TM^ RNA extraction kit (Thermo Fisher, 15596026, Waltham, MA, USA). The concentration of the RNA from each sample was determined using a NanoDrop 2000 (Thermo Scientific, MA, USA), and the integrity of the RNA was determined using gel electrophoresis. The total RNA was subsequently reverse-transcribed to quantify the desired cDNA using the PrimeScript^TM^ RT reagent kit reverse transcription kit (TaKaRa, Tokyo, Japan) according to the instructions. The RT–qPCR was performed using an ABI QuantStudio 3-sequence detection system (Applied Biosystems, Waltham, MA, USA) using the SYBR Premix Ex Taq Kit (TaKaRa, Tokyo, Japan) with the following reaction conditions: 95 °C for 3 min, 40 cycles of 95 °C for 15 s, 56 °C for 30 s, and 72 °C for 20 s. The reference gene TCTB was used for normalization, the expression data were calculated using the 2∆∆Ct formula, and the difference was calculated using the SPSS 20.0 software (LSD method, *p* < 0.05). The primers used for quantification are listed in Appendix A.

## Figures and Tables

**Figure 1 plants-12-04079-f001:**
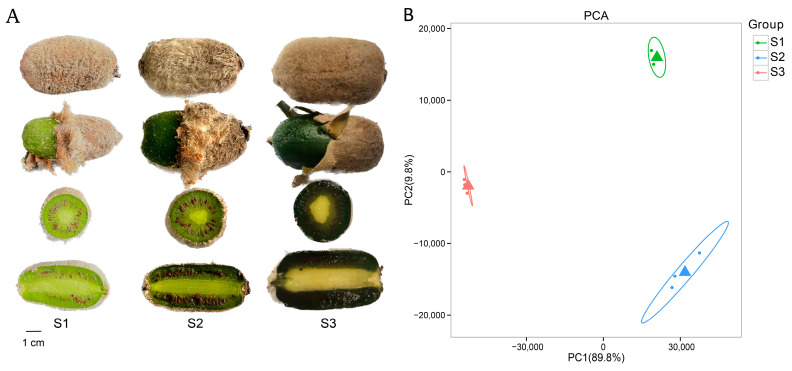
(**A**) The three developmental stages of *A. eriantha* fruit, S1: fruit at 180 days after flowering, S2: fruit at 210 days after flowering, S3: fruit at 240 days after flowering. (**B**) Principal component analysis of three fruit samples based on gene expression profiles.

**Figure 2 plants-12-04079-f002:**
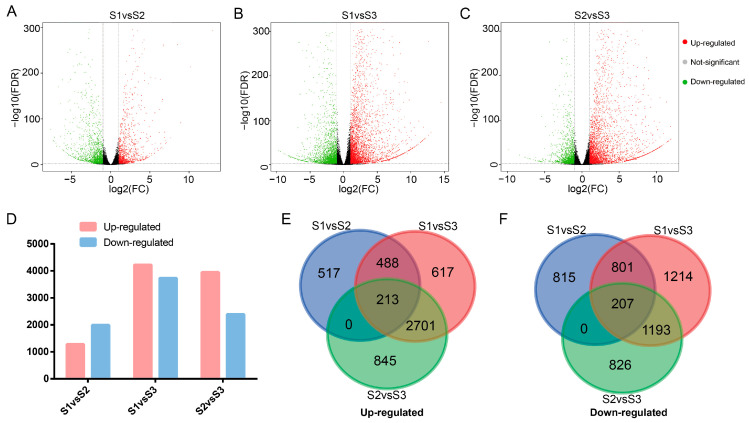
DEGs during *A. eriantha* maturation. (**A**–**C**) Volcano plot of DEGs between the three stages of *A. eriantha* ripening. (**D**) Number of DEGs. (**E**,**F**) Statistics of the core upregulated and downregulated DEGs.

**Figure 3 plants-12-04079-f003:**
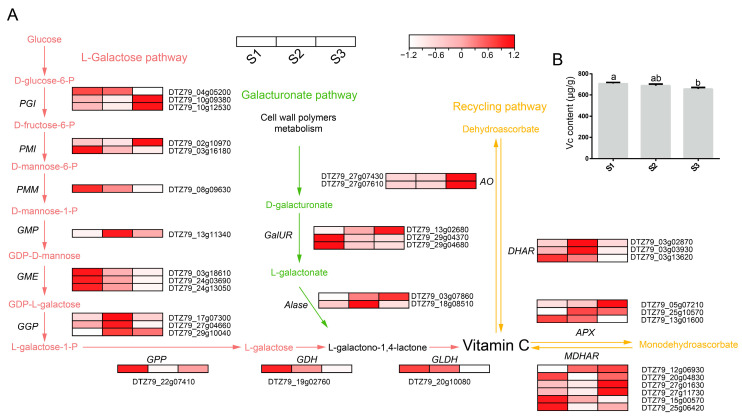
Schematic diagram of the *A. eriantha* Vc metabolic pathway. (**A**) Metabolite and gene expression profiles of Vc synthesis in *A. eriantha*. *AO*: L-ascorbate oxidase; *APX*: L-ascorbate peroxidase; *DHAR*: dehydroascorbate reductase; *GDH*: L-galactose dehydrogenase; *GGP*: GDP-L-galactose phosphorylase; *GLDH*: L-galactono-1,4-lactone dehydrogenase; *GME*: GDP-D-mannose-3,5-epimerase; *GMP*: GDP-D-mannose pyrophosphorylase; *GPP*: L-galactose-1-phosphate phosphatase; *MDHAR*: monodehydroascorbate reductase; *PGI*: glucose-6-phosphate isomerase; *PMI*: mannose-6-phosphate isomerase; *PMM*: phosphomannomutase, *Alase*: aldonolactonase; *GalUR*: D-galacturonate reductase. Blue arrows indicate the L-galactose pathway, green arrows indicate the galacturonate pathway, and yellow arrows indicate the recycling pathway. (**B**) Determination of Vc content during *A. eriantha* development. Differential analysis was performed using the LSD method with *p* < 0.05.

**Figure 4 plants-12-04079-f004:**
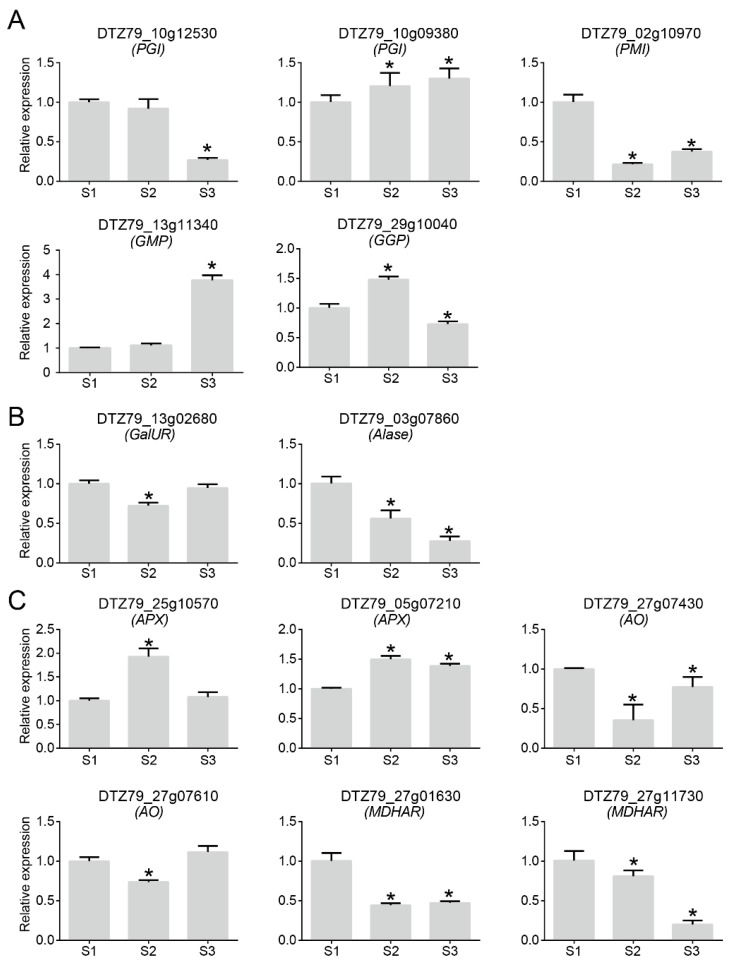
Analysis of expression patterns of related genes in different Vc synthesis pathways in *A. eriantha*. (**A**) L-galactose pathway, (**B**) galacturonate pathway, (**C**) recycling pathway. Asterisks indicate significant difference.

**Figure 5 plants-12-04079-f005:**
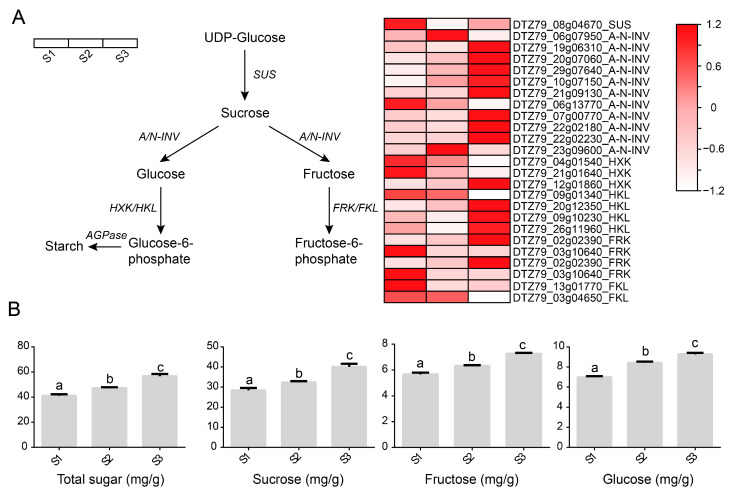
Schematic diagram of the *A. eriantha* glucose metabolism pathway. (**A**) Pathway of sucrose, fructose, and glucose metabolism in *A. eriantha*. *SUS*: sucrose synthase, *A/N-INVc*: invertase, *HXK*: hexokinase, *FRK*: fructokinase, *AGPase*: ADP-glucose pyrophosphorylase, *HKL*: hexokinase-like, *FKL*: fructokinase-like. (**B**) Determination of total sugar, sucrose, fructose, and glucose contents in *A. eriantha*. Letters indicate significant difference.

**Figure 6 plants-12-04079-f006:**
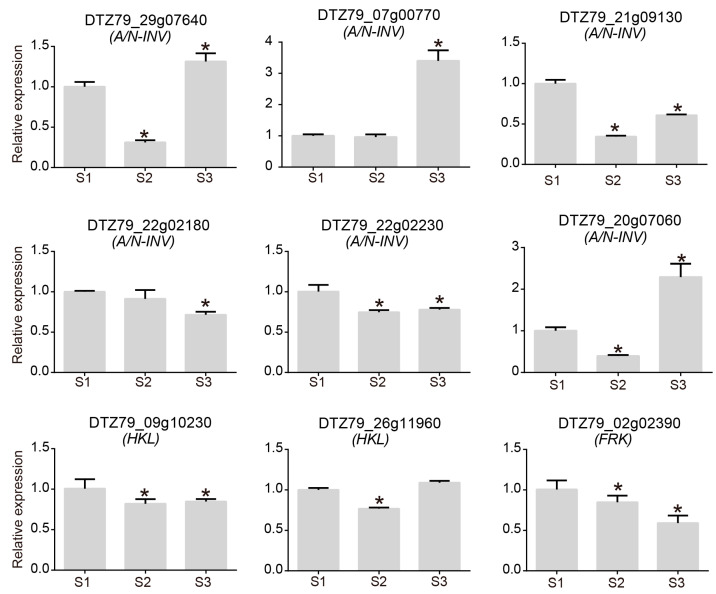
Quantitative analysis of genes related to glucose metabolism in *A. eriantha*. Asterisks indicate significant difference.

**Figure 7 plants-12-04079-f007:**
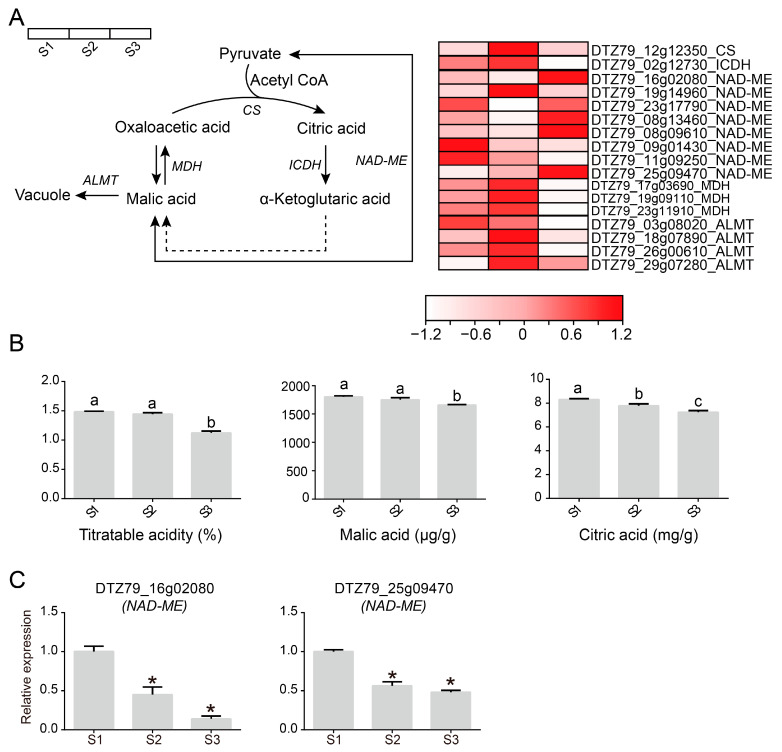
Schematic diagram of organic acid metabolism pathways in *A. eriantha*. (**A**) Metabolic pathways of malic acid and citric acid in *A. eriantha*. *CS*: citrate synthase, ICDH: isocitrate dehydrogenase, *ALMT*: AL-activated malate transporter, *MDH*: malate dehydrogenase, *NAD-ME*: malic enzyme. (**B**) Determination of titratable acid, malic acid, and citric acid in *A. eriantha*. Letters indicate significant difference. (**C**) Analysis of the expression patterns of two *NAD-ME* genes during the maturation process of *A. eriantha*. Asterisks indicate significant difference.

## Data Availability

All sequencing data, including those of the genome and transcriptome, can be found in the China National Center for Bioinformation database (PRJCA014460).

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
