# Peer review of "Transcriptome Analysis Reveals Fruit Quality Formation in *Actinidia eriantha* Benth"

_plants, 2023, doi:10.3390/plants12244079_

Round 1
Reviewer 1 Report (New Reviewer)
Comments and Suggestions for Authors
Line 17 Therefore, we sequenced the transcriptome of A. eriantha fruits at different developmental stages. What is the characteristic of the fruit of A. eriantha? Color, shape etc.
Line 18-19-20 is not clear!
Line 67—71 How much vitamin C, sugar, ascorbic acid, etc. are they present in the fruit of A. eriantha?
Line 74-75 delete.
line 83 A. eriantha is a unique wild germplasm resource of kiwifruit in China, and its fruits are rich in Vc, minerals and a variety of amino acids, with unique flavors.
How many accessions of A. eriantha have been recorded in China?
Line 92 Add bibliography:
Huimin Jia et al., 2022 on A. eriantha nutritional component
D.Lei et al., 2022 which provides information on the variations in nutritional compounds during the ripening period of the kiwi and on the mechanism of sugar accumulation.
Line 121 Table 4 is missing.
Line 124 Table 5 is missing.
Line 259 and line 262 A. chinensis “Hort16A”, I suppose!
Line 313 “Vc is indispensable nutrient for humans.” The sentence is unneeded
Line 119-124 The authors state that GO enrichment analysis indicates an important role in hormone accumulation and metabolism during fruit ripening,
but they do not discuss these genes.
The authors says: “This study provides a reference for understanding the formation of unique flavors of A. eriantha, which can be used for genetic breeding and variety improvemen”. And then, “This study utilized high-throughput sequencing and molecular biology experiments to preliminarily explore the genes of A. eriantha and their function in Vc synthesis, providing valuable information for molecular breeding and germplasm resource development of kiwifruit and laying the foundation for genetic research on A. eriantha”.
Could the author highlight better how this work could be useful in the breeding process?
Comments on the Quality of English Language The quality of the language and English used in this article is good. no revisions are necessary.
Author Response
Dear Editor and Reviewers,
Thank you for Editor’ and Reviewers’ suggestions and comments. These comments are very helpful to improve our manuscript. We have carefully revised each comment. Modified areas have been marked with yellow in the manuscript. Please see the attachment.

Reviewer 2 Report (New Reviewer)
Comments and Suggestions for Authors
By using high-throughput sequencing and molecular biology experiments to explore the genes involved in Vitamin C synthesis, the study provides useful basis for molecular breeding of kiwifruit. Overall, the study is useful following a transcriptomic and biochemical approach to mine the genetic basis of Vc synthesis. Each section is nicely written and presented.
With a few minor errors in the writing, I'd suggest to have a thorough read of the whole manuscript to avoid them.
Comments on the Quality of English Language
A few minor typos and grammatical error are present which could be revised by having a thorough read of the whole manuscript text.
Author Response
Dear Editor and Reviewer,
Thank you for your comments, we have invited colleagues of native English speakers to make English modifications, the typo in the manuscript was also corrected.
Reviewer 3 Report (New Reviewer)
Comments and Suggestions for Authors
Dear authors,
Many thanks for this interesting piece of knowledge about fruit quality formation on A. eriantha. The study aimed to systematically analyze the factors affecting the quality and flavour composition of A. eriantha, specifically vitamin C (Vc), sugar and organic acids. Transcriptome sequencing was performed at different developmental stages of A. eriantha, resulting in the identification of upregulated and downregulated differentially expressed genes (DEGs). The upregulated DEGs were associated with hormones, sugars, organic acids, and various organic metabolic pathways, while the downregulated DEGs were related to the light signalling pathway. The article also presents the metabolic pathways of sugars, organic acids, and Vc in A. eriantha and identifies genes involved in their synthesis at different fruit development stages. The study found that Vc content decreased, carbohydrate compound content increased, and organic acid content decreased during fruit development, with gene expression patterns closely linked to the accumulation patterns of these compounds. The results provide insights into Vc, sugar, and organic acid accumulation in A. eriantha and contribute to understanding flavour formation in this fruit.
Please, find my following suggestion for the revision of your publication:
Abstract:
Do not use abbreviations in the abstract such as Vc, DEGs or GO without explaining the meaning. If included, describe the meaning again in the following section when appears for the first time.
Introduction:
The description of the acquisition of fruit quality traits in such detail must be fully revised. I highly recommend just a general overview or state of art of fruit quality research at the moment in A. eriantha and related species and clarifying research objectives starting by summarizing the research objectives clearly. Highlight the main purpose of the study, which is to understand the factors influencing the quality and flavour of A. eriantha, specifically vitamin C, sugar, and organic acids; why it is important and possible applications. The references in the text must be improved including more transcriptome analysis during fruit ripening in woody plants as:
Identification of QTLs linked to fruit quality traits in apricot (Prunus armeniaca L.) and biological validation through gene expression analysis using qPCR. BE García-Gómez, JA Salazar, L Dondini… - Molecular Breeding, 2019 or Analysis of metabolites and gene expression changes relative to apricot (Prunus armeniaca L.) fruit quality during development and ripening BE García-Gómez, D Ruiz, JA Salazar, M Rubio… - Frontiers in plant science, 2020
Blueberry, watermelon, banana and rosa are not considered fruit trees.
Line 98. The verification of gene expression and inclusion is not well described. What do you mean with inclusions? Are RNA-Seq DEGs verified by RT-qPCR? Did you find any relation between gene expression and metabolites?
Fruit varieties' names must be written between '', scientific species names in italics and common names do not start with capital letters. Try to follow a consistent nomenclature along the text referring to kiwifruit (not kiwi or kiwi fruit).
Results:
In trancriptomic analysis, describe the meaning of Q30 for sequencing quality and the threshold set for filtering low-quality.
In the PCA, the percentage represented in the axes is the percentage of explained variance (PEV) for each of the principal components. Indicate the meaning of PCA and PC.
Line 149: Sentences "To further elucidate the correlation between Vc synthesis and metabolism and genes in A. eriantha" seems incomplete.
Line 150. Inconsistent gene names. Indicate gene IDs. Gene names must be written in capital letters and italics along the manuscripts.
To validate the results obtained in both RNA-Seq and RT-qPCR the genes must show the same expression trend between the two methods. Represent this data together estimating the correlation between results. Seems to be inconsistent results for genes such as PGI, PMI, GalUR, Alase, etc. This methodology must be applied along all the metabolic pathways assays (sugars and acids), including Vc. Besides, include a correlation matrix between all the metabolites and genes analyzed.
Discussion:
Line 273. "Our results show that the genes involved in sugar and organic acid synthesis during fruit development of A. eriantha have a highly similar expression trend with related sugars and organic acids, suggesting that sugars and organic acids are regulated by a complex network." This affirmation is not well supported by the results as there is a lack of statistical analysis that could verify this as the 'similarity' is not observed.
Materials and methods:
The sampling methodology and RNA extraction must be described in detail. Describe the differential development stages (S1, S2 and S3), the decision of sampling at those moments, the tissue sampled and how many biological replicates were taken. Additionally, describe the kiwifruit variety selected for sequencing.
Include references and versions for all software used in data analyses and kiwifruit genome. Describe the software used in functional enrichment analyses and channel (maybe you mean pathway?) significance enrichment.
Specified which assay kits were used for each metabolite content determination and the sample preparation and methodology followed. Review SYBR kit reference and naming.
Author Contributions: In the article, there were 8 authors but only 4 participated in the design, experiment performing, data analysis and writing. Indicate the contribution of the rest of the authors.
Supplementary material: Headers of supplementary material not written in English.
Comments on the Quality of English Language
After carefully reading the pre-print of this article, I make the following suggestion to improve the text:
The general structure of some sections must be reviewed and modified.
The vocabulary must be improved to sound more specific.
Author Response
Dear Editor and Reviewers,
Thank you for Editor’ and Reviewers’ suggestions and comments. These comments are very helpful to improve our manuscript. We have carefully revised each comment. Modified areas have been marked with yellow in the manuscript. Please see the attachment.

Round 2
Reviewer 3 Report (New Reviewer)
Comments and Suggestions for Authors
Dear authors,
Congratulations on the many suggestions you included in the revision of the manuscripts but some issues must be corrected before publication. Some of the most important that were not satisfied:
Abbreviations: The first time you refer to the whole explanation and the abbreviation must be between ()
Related to the bibliography: In the methodology, the software citation doesn't follow the format of the publication. In addition, review the format of all the references as many errors were detected.
Results: Still no correlation between gene expression and metabolite content. If the results obtained in the RNA-Seq do not correlate with the genes expected to be expressed, it must be indicated clearly in the manuscript and try to justify why you use other genes from metabolic pathways to validate your hypothesis.
Best,
Reviewer
Author Response
Thank you for Editor’ and Reviewers’ suggestions and comments. These comments are very helpful to improve our manuscript. We have carefully revised each comment. Modified areas have been marked with yellow in the manuscript. Here is our response to reviewer,please see the attachment.

This manuscript is a resubmission of an earlier submission. The following is a list of the peer review reports and author responses from that submission.
Round 1
Reviewer 1 Report
Comments and Suggestions for Authors
The study of genetic regulation is extremely interesting work, particularly this document shows us many situations on the activation and repression of genes involved in the growth and development of kiwi fruits. These characteristics are definitely related to the quality of the fruit.
It was very interesting to read this work and I am sending the draft attached for some minor observations.

Author Response
Dear reviewers:
Thank you for your approval of our submitted article. First of all, I strongly accept your comments on this article. We have revised them one by one and marked them in the text in red and blue fonts. At the same time, I will also submit the revised draft to the system. Thanks again for your hard work.good luck.

Reviewer 2 Report
Comments and Suggestions for Authors
Your research and paper seems straightforward--a clear correspondence among your introduction's four paragraphs and your research results and your discussion and your methods is established. My internal comments (on attached manuscript copy) show that I was indeed able to follow your logic well (although I did request some clarifications at several points). My main concern is a substantive one, however--at a high level (across plant physiology disciplines), what main insights has your work developed? What might it mean to someone who is working with apples (or carrots, or pines)? In other words, this manuscript actually feels like a fairly predictable result (in terms of the genes up- and down-regulated and the enzymatic pathways amplified or quieted). Can you draw any general inferences from this work, beyond the narrow range of this suite of related species? Those thoughts could come into your introduction, and could especially enter into your discussion (and conclusions).

Comments on the Quality of English Language
The only English concern that I have is about the way that you talk about consumers "loving" this fruit. Seems to be a bit over-generalized and a bit unscientific.
Author Response

(The authors gave the same response as above.)

Round 2
Reviewer 2 Report
Comments and Suggestions for Authors
Thank you for your responses to the reviews of your manuscript. On a technical level, you have addressed all of this reviewer's technical concerns. I remain concerned on the level of generalizability; perhaps, that should be described as a philosophical concern. If you recall, I wondered if you could address, via your introduction and discussion/conclusions, the applicability of your work to other related plant contexts. I could not find evidence that this request was addressed. Perhaps the editor concluded that my request was not relevant?